



# Venus's induced magnetosphere during active solar wind conditions at BepiColombo's Venus 1 flyby

Martin Volwerk[1], Beatriz Sánchez-Cano[2], Daniel Heyner[3], Sae Aizawa[4], Nicolas André[4], Ali Varsani[1], Johannes Mieth[3], Wolfgang Baumjohann[1], Richard Harrison[8], Harald Jeszenszky[1], David Fischer[1], Yoshifumi Futaana[9], Iwai Kazumasa[5], Gunter Laky[1], Yoshizumi Miyoshi[5], Rumi Nakamura[1], Ferdinand Plaschke[1], Ingo Richter[3], Sebastián Rojas Mata[9], Yoshifumi Saito[7], Daniel Schmid[1], Daikou Shiota[6], and Cyril Simon Wedlund[1]

[1]Space Research Institute, Austrian Academy of Sciences, Graz, AT
[2]School of Physics and Astronomy, University of Leicester, Leicester, UK
[3]Institute for Geophyscs and Extraterrestrial Physics, Braunschweig Institute of Technology, DE
[4]IRAP, CNRS-UPS-CNES, Toulouse, France
[5]Institute for Space-Earth Environmental Research, Nagoya University, Japan
[6]National Institute of Information and Communications Technology, Tokyo, Japan
[7]Institute of Space and Astronautical Science, Japan Aerospace Exploration Agency, Kanagawa, Japan
[8]RAL Space, UKRI-STFC Rutherford Appleton Laboratory, Harwell Campus, Oxfordshire, UK
[9]Swedish Insitute of Space Physics, Kiruna, Sweden

**Correspondence:** MARTIN VOLWERK (martin.volwerk@oeaw.ac.at)

**Abstract.** Out of the two Venus flybys that BepiColombo uses as a gravity assist manoeuvre to finally arrive at Mercury, the first took place on 15 October 2020. After passing the bow shock, the spacecraft travelled along the induced magnetotail, crossing it mainly in the $Y_{VSO}$-direction. In this paper, the BepiColombo Mercury Planetary Orbiter Magnetometer (MPO-MAG) data are discussed, with support from three other plasma instruments: the Planetary Ion Camera (PICAM), the Mercury

Electron Analyser (MEA) and the radiation monitor (BERM). Behind the bow shock crossing, the magnetic field showed a draping pattern consistent with field lines connected to the interplanetary magnetic field wrapping around the planet. This flyby showed a highly active magnetotail, with, e.g., strong flapping motions at a period of $\sim 7$ min. This activity was driven by solar wind conditions. Just before this flyby, Venus's induced magnetosphere was impacted by a stealth coronal mass ejection, of which the trailing side was still interacting with it during the flyby. This flyby is a unique opportunity to study the full length

and structure of the induced magnetotail of Venus, indicating that the tail was most likely still present at about 48 Venus radii.

## 1 Introduction

The interaction of Venus with the magnetoplasma of the solar wind gives rise to the creation of a so-called induced magnetosphere (see e.g., Luhmann et al., 1986; Phillips and McComas, 1991; Bertucci et al., 2011; Dubinin et al., 2011; Futaana





et al., 2017). The solar wind is first braked by the upstream bow shock and is then further mass-loaded and slowed down due to the ionization of exospheric particles and their pick-up by the solar wind convection electric field, whilst approaching the planet. The magnetic field is subsequently draped around the planet (e.g. Saunders and Russell, 1986) in what is often called a comet-like interaction.

Closer to the planet the magnetic field piles up in a region that is known under various names: magnetic pile-up boundary,
magnetic barrier or magnetopause (Zhang et al., 2008b, a). In this region, the interplanetary magnetic field (IMF) is stopped at the sunward side of the planet and cannot penetrate into the ionosphere. This boundary extends downstream to at least 11 planetary radii and encloses the induced magnetotail where planetary plasma escape mainly occurs (Bertucci et al., 2011). One more boundary is created through the difference in plasma composition, where there is a strong gradient in the energetic electrons and the ion population starts to become dominated by planetary ions instead of solar wind ions (Martinecz et al.,
2009$a$, $b$), the ion composition boundary. Finally, an additional boundary related to the upper limit of the collisional ionosphere is typically found at lower altitudes, the ionopause. This boundary is where the thermal ionospheric pressure balances the induced magnetosphere's magnetic pressure (Bertucci et al., 2011); it occurs mainly on the dayside and post-terminator nightside sectors.

In the dayside and upstream region of the induced magnetosphere various kinds of plasma waves are typically detected. In
particular, two wave modes related to the pick-up of freshly created ions (Gary, 1992) in Venus's exosphere play an important role. In the solar wind, proton cyclotron waves are observed (Delva et al., 2008, 2015) created by the ion pick-up in a relatively low plasma-$\beta$ environment. Behind the quasi-perpendicular bow shock, mirror-modes are often found (Volwerk et al., 2008$a$, $b$, 2016) because of the relatively high plasma-$\beta$ there and the mainly perpendicular-to-the-magnetic-field energization of the ions crossing the bow shock.

In Venus's downstream region the induced magnetotail is created by the draped field lines, producing two regions of oppositely directed magnetic field separated by a current sheet (Phillips and McComas, 1991), not unlike the Earth's magnetotail. The direction of the field in the tail is mainly aligned with the direction of the solar wind and the field in the lobes is stronger than that in the magnetosheath (Russell et al., 1981). A difference in wave power between the magnetosheath and the tail proper can also be seen (Russell et al., 1981; Vörös et al., 2008a, b). As in the Earth's magnetotail, magnetic reconnection has been
observed to take place (Volwerk et al., 2009, 2010; Zhang et al., 2010).

The first flythrough of Venus's magnetotail was done by Mariner 10 (Lepping and Behannon, 1978), from as far downstream as $\sim 100\,R_V$. The induced magnetosphere of Venus has only been studied over a limited region of space, because of the limited orbital coverage of the visiting spacecraft. Pioneer Venus Orbiter (PVO) did not explore the central region of the tail further than $\sim 11.5\,R_V$ downstream of Venus, and Venus Express (VEX), due to a larger inclination of the spacecraft orbit, did not
venture beyond $\sim 4\,R_V$ downstream. This means that the structure and the dynamics of the Venusian far tail have not been fully charaterised yet. Important questions are still open with respect to, e.g., the length of the tail and bow shock/wave along it: where does it "merge" with the ambient solar wind? How do flux ropes and plasmoids move through the far tail? Learning this will have strong implications for understanding the processes that encourage the atmosphere to escape or shield it from doing so..





Recently, however, three newly launched missions have performed flybys using Venus as a gravitational assist to get into the correct orbit towards the inner solar system.

The first one was Parker Solar Probe (PSP, Fox et al., 2016), which is set to use 7 Venus flybys to adjust its perihelion distance. The first flyby was on 3 October 2018, the second on 26 December 2019 approaching from the downstream direction, and the third on 11 July 2020, approaching from the upstream direction. The first flyby passed into the induced magnetosphere,

where strong kinetic-scale turbulence was found in the magnetosheath (Bowen et al., 2021) as well as sub-proton scale magnetic holes (Goodrich et al., 2021), whereas the second flyby grazed Venus's bow shock at the dawn terminator and double layers were observed at this boundary (Malaspina et al., 2020).

BepiColombo is the second new mission with two planned Venus flybys (Benkhoff et al., 2010; Milillo et al., 2020; Mangano et al., 2021), the first of which is the topic of this paper. The third mission is Solar Orbiter (Müller et al., 2013, 2020), which

had its first Venus flyby about 2 months after the first BepiColombo flyby, on 27 December 2020.

This paper focuses on the first BepiColombo flyby that occurred on 15 October 2020. Since this flyby was the first opportunity to have scientific planetary observations after the instrumental tests performed during the Earth flyby on 10 April 2020, several science instruments were turned on for this planetary encounter. The BepiColombo trajectory was such that by making a long transit into the Venusian induced magnetotail, it allowed for a precious opportunity to study the dynamics and structures

of the tail, including the far tail, a region mostly unexplored.

## 2    The Data

The first BepiColombo flyby occurred on 15 October 2020 with closest approach at 03:58:31 UT and a minimum altitude of 10720.5 km above the planet surface ($\sim 2$ Venus radii). BepiColombo was in the solar wind and crossed the Venusian bow shock on the day side in the evening sector, and then it did a long transit into the induced magnetotail. The flyby is shown

in Fig. 1 in the Venus solar orbital (VSO) coordinate system. In this figure, the Sun is to the left ($+X_{\mathrm{VSO}}$), and the different plasma boundaries together with BepiColombo's trajectory are indicated.

We use data from the BepiColombo (Anselmi and Scoon, 2001; Benkhoff et al., 2010) magnetometer MPO-MAG on board the Mercury Planetary Orbiter (MPO) spacecraft (Glassmeier et al., 2010; Heyner et al., 2021), at a cadence of 1 second (Fig. 2) and a low-pass filter for periods below 5 minutes (Fig. 3) in order to get the large-scale structure of the induced magnetosphere

undisturbed by high-frequency oscillations. We limit the discussion of the observations to the interval of 04:14 UT (crossing of the bow shock) to 12:00 UT, spannning the region of $\sim 0 \geq X_{\mathrm{VSO}} \geq -40\,\mathrm{R_V}$ (Venus radius, $R_{\mathrm{V}} = 6052\,\mathrm{km}$).

This work focuses on different regions within the induced magnetosphere that are marked with purple-, green-, blue- and red-colours along the trajectory in Fig. 1.

In order to interpret the structure of the induced magnetosphere, the cone ($\theta_c$) and clock ($\phi_c$) angle of the magnetic field are

calculated:



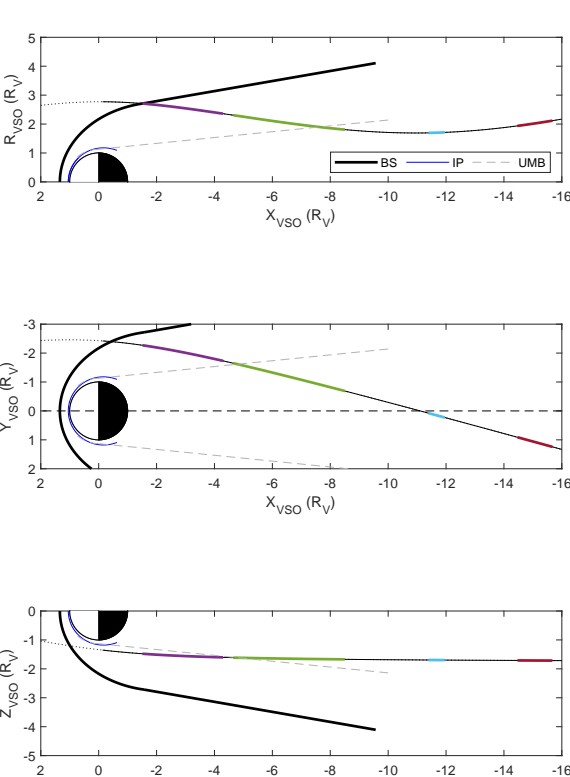

**Figure 1.** The BepiColombo first flyby to Venus in VSO coordinates (with $R_{\mathrm{VSO}} = \sqrt{Y_{\mathrm{VSO}}^2 + Z_{\mathrm{VSO}}^2}$). The thick black line is the bow shock (BS) for solar minimum conditions (Zhang et al., 2008b), the thin blue line is the ionopause (IP) (Zhang et al., 2008b) and the grey dashed lines are the upper mantle boundary (UMB) (Martinecz et al., 2009b). The thin black (dotted) line is the orbit of BepiColombo, with the solid line showing the interval discussed in this paper. The purple, green, blue and red marked intervals are of special interest.



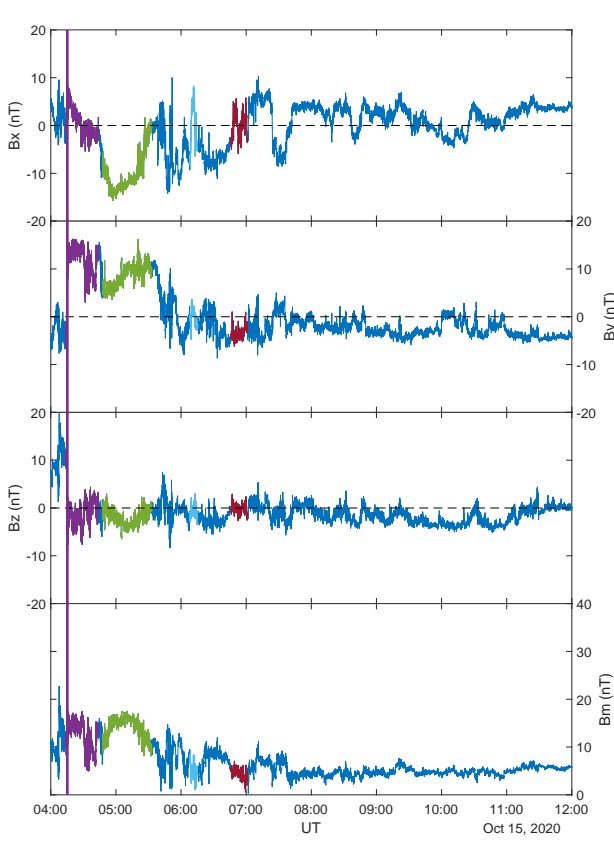

**Figure 2.** Full 1 s. resolution MPO-MAG data. Top to bottom panels show the $B_{\mathrm{x}}, B_{\mathrm{y}}, B_{\mathrm{z}}$ and the absolute magnetic ($B_{\mathrm{m}}$) field components, respectively. The vertical purple line marks the bow shock transit.



$$\theta_{\mathrm{c}} = \tan^{-1}\left(\frac{\sqrt{B_{\mathrm{y}}^2 + B_{\mathrm{z}}^2}}{B_{\mathrm{x}}}\right), \tag{1}$$

$$\phi_{\mathrm{c}} = \tan^{-1}\left(\frac{B_{\mathrm{z}}}{B_{\mathrm{y}}}\right). \tag{2}$$

These two angles describe the direction of the field: a cone angle of $\theta_{\mathrm{c}} = 0°/180°$ indicates an sunward/anti-sunward direction and $\theta_{\mathrm{c}} = 90°$ indicates a field direction perpendicular to the Venus-Sun line. The clock angle shows the direction in the plane perpendicular to the Venus-Sun line with $\phi_{\mathrm{c}} = 0°/90°$ indicating a field in the $Y_{\mathrm{VSO}}/Z_{\mathrm{VSO}}$ direction. In Fig. 3 the magnetometer data are shown, as well at the cone and clock angles and the location of the spacecraft.

Data from Planetary Ion Camera (PICAM), part of the SERENA (Search for Exospheric Refilling and Emitted Natural Abundances) instrument suite (Orsini et al., 2010, 2021*a*, *b*), are also used to support the magnetometer data. PICAM is an ion mass spectrometer, which operates as an all-sky camera for charged particles. It is optimised for Mercury's observations, to study the chain of processes by which neutrals are ejected from Mercury's soil, and are eventually ionised and transported through the Hermean environment. PICAM operates by scanning through the energy and angular distribution of ions effectively from 10 eV up to 3 keV, and with a field of view of $1.5\pi$ sr and a cadence of 64 s. PICAM also provides ion composition for a mass range extending up to $\sim 132$ u (Xenon).

Electron data from the Mercury Plasma Particle Experiment (Saito et al., 2010; Saito and et al., 2021) on board the Mercury Magnetospheric Orbiter (MMO, renamed Mio after launch) spacecraft of BepiColombo are also utilised. In particular, data from the Mercury Electron Analyzer (MEA) 1 in solar wind mode (3 eV - 3000 eV) are used to investigate the low-energy electron distribution during the flyby at a cadence of 4 s. Since the MMO spacecraft is stuck behind the MOSIF sunshield during cruise phase, MEA1 has a limited field of view but, despite this, useful scientific observations can be obtained since low-energy electrons are almost isotropic.

In order to account for the solar wind activity responsible for the IMF disturbances around the Venus 1 flyby, data from the BepiColombo Radiation Monitor (BERM) are used. BERM is a particle detector able to provide radiation information, in a way similar to the Standard Radiation Environment Monitor (SREM) instrument aboard several ESA missions such as Rosetta (Honig et al., 2019). In particular, it is able to measure high-energy charged particles (e.g., electrons and protons), and the higher energy channels background counts can be used as a proxy for galactic cosmic rays.

Moreover, we also use data from the Large Angle and Spectrometric Coronagraph (LASCO) instrument on board the Solar and Heliospheric Observatory satellite (SOHO) (Brueckner et al., 1995). In particular, we use data from the c2 white light coronagraph imaging from 1.5 to 6 solar radii. We also use the Heliospheric Imager (HI) instrument, that forms part of the Sun Earth Connection Coronal and Heliospheric Investigation (SECCHI) suite of remote sensing instruments on board the Solar TErrestrial RElations Observatory (STEREO)-A spacecraft. The HI is a wide-angle visible-light imaging system for the detection of coronal mass ejection (CME) events in interplanetary space covering the region of the heliosphere from 4 degrees to 88 degrees elongation measured from the Sun-centre (Howard et al., 2008; Eyles et al., 2009). It consists of two telescopes, HI1 and HI2: in this study we have used only images from HI1.





**Table 1.** Selected time intervals, based on the magnetometer data, showing different regions in Venus's induced magnetosphere behind the bow shock. The distance in the tail behind Venus in $X_{\mathrm{VSO}}$ is given in Venus radii, $R_{\mathrm{V}}$.

| region | time in | time out | distance $|X_{\mathrm{VSO}}|$ | box colour |
|---|---|---|---|---|
| bow shock & magnetosheath | 04:14 | 04:44 | $1.5 - 4.2$ | purple |
| magnetotail | 04:48 | 05:33 | $4.2 - 8.5$ | green |
| around neutral sheet | 05:23 | 06:08 | $9.0 - 15.7$ | NA |
| neutral sheet crossings | 06:08 | 06:15 | $11.3 - 12.0$ | blue |
| flapping region | 06:46 | 07:01 | $14.5 - 15.7$ | red |
| magnetotail | 07:45 | 14:00 (?) | $15.7 - 48$ (?) | NA |

Finally, we have also used the Space-weather-forecast-Usable System Anchored by Numerical Operations and Observations (SUSANOO) model from Nagoya University to simulate the solar wind conditions encountered by BepiColombo at Venus dur-

ing the flyby (Shiota et al., 2014; Shiota and Kataoka, 2016). SUSANOO is a magnetohydrodynamic (MHD) solar wind model of the inner heliosphere between $25$ and $425$ solar radii using a Yin-Yang grid, where the velocity, density and temperature are obtained from empirical models of the solar wind (Odstrčil and Pizzo, 1999*a, b*). CMEs are included in the inner boundary of the simulation as spheromak-type magnetic flux ropes (Shiota et al., 2014; Shiota and Kataoka, 2016; Iwai et al., 2019) with initial velocities derived semi-automatically from SOHO-LASCO.

In Fig. 3 there are four regions marked by differently coloured vertical lines, which will be discussed in more detail below. These intervals are also marked along the orbit of the flyby in Fig. 1. The times when these regions were transited and the distance to the planet when they occurred are listed in Table 1.

## 3 MPO-MAG Observations

First the MPO-MAG data, based on the different regions as listed in Table 1 will be discussed.

### 3.1 Magnetosheath Draping

After crossing the bow shock at $\sim 04:14$ UT the spacecraft enters the Venusian magnetosheath. Fig. 4 shows a zoom-in on the field in the magnetosheath. It is clear that after the crossing of the bow shock (the first purple vertical line), the magnetic field rotates strongly from $B_{\mathrm{z}}$ (yellow) into $B_{\mathrm{y}}$ (red), which is also evident from the clock angle, $\phi_{\mathrm{c}}$, that turns from $\sim 90°$ to $\sim 0°$. $B_{\mathrm{x}}$ is the minor component in this interval, as can be clearly seen in the cone angle, $\theta_{\mathrm{c}} \sim 90°$.

This means that, in the magnetosheath, the magnetic field is mainly in the $Y_{\mathrm{VSO}}$-direction, i.e., perpendicular to the induced magnetotail direction. This is reminiscent of the pattern described by Delva et al. (2017, their figure 1) where draped magnetic field lines in the magnetosheath were connected to the IMF, albeit that BepiColombo makes a much further excursion away

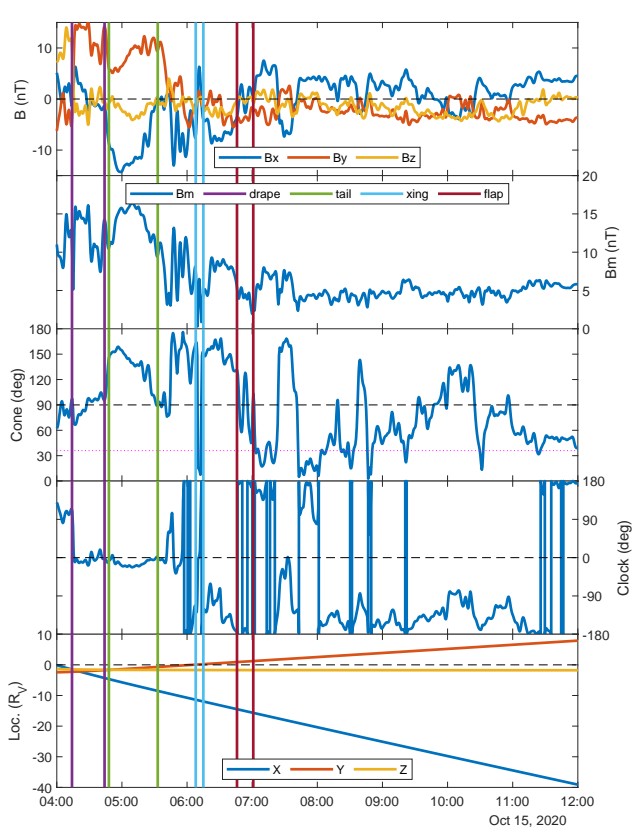

**Figure 3.** Magnetometer data in the magnetosheath and tail. From top to bottom: the three components of the magnetic field in VSO-coordinates; the magnitude of the magnetic field; the cone angle; the clock angle; and the location of the spacecraft in VSO-coordinates. The purple, green, blue and red dotted vertical lines show the intervals of interest.



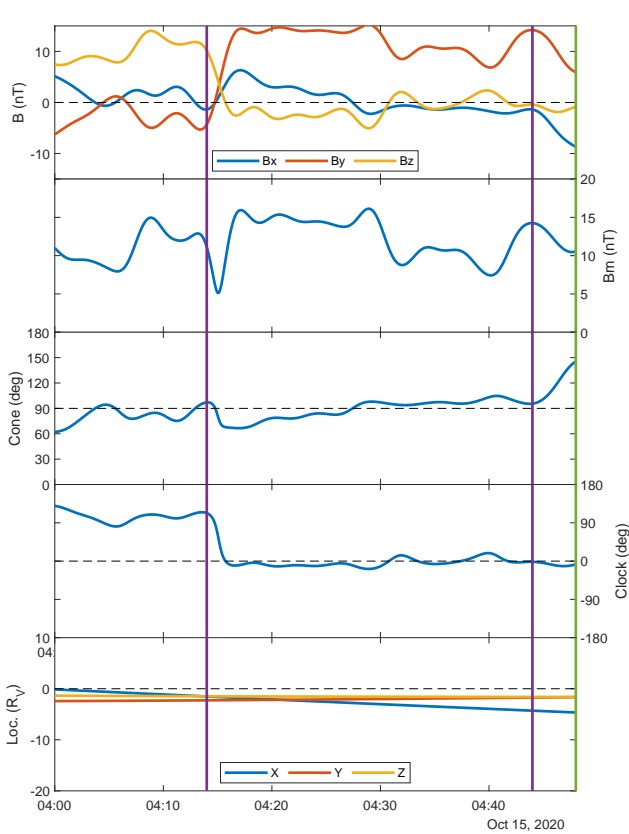

**Figure 4.** Zoom in on the magnetosheath interval (purple) where the cone angle $\theta_c \approx 90°$ and the clock angle $\phi_c \approx 0°$. This indicates that the magnetic field is pointing in the $Y_{\text{VSO}}$-direction.





from Venus, in this interval up to $X_{\mathrm{VSO}} \approx -4\,R_{\mathrm{V}}$, than VEX. This draping pattern was shown to exist in hybrid plasma simulations by Jarvinen et al. (2013).

### 3.2 Magnetotail Draping

After passing through the magnetosheath, there is a strong rotation of the magnetic field, at $\sim 04:44$ UT, where $B_{\mathrm{y}}$ decreases and $B_{\mathrm{x}}$ increases and the cone angle changes from $\theta_{\mathrm{c}} \approx 90°$ to $\approx 150°$ as seen in Fig. 5 between the second purple and first green vertical line. Here, the magnetic field takes on the shape of a magnetotail, with the main direction along the Venus-Sun direction, albeit with a significant $B_{\mathrm{y}}$ contribution.

Because of the conic shape of the bow shock behind Venus, the magnetic field in the magnetosheath and magnetotail is not strictly along the Venus-Sun line, but flares out following this conic shape. A significant $B_{\mathrm{y}}$ contribution can be caused by this flaring of the magnetotail. However, we see in Fig. 5 that $B_{\mathrm{x}} < 0$ and $B_{\mathrm{Y}} > 0$, which is incompatible with flaring, for which one would expect $B_{\mathrm{Y}} < 0$. This means that the "cross-tail magnetic field" $B_{\mathrm{y}}$ needs to have its origin elsewhere, e.g. from penetrating IMF into the tail. This can be caused via reconnection of the induced magnetic field with IMF structures. This process is well known from Earth (e.g., Fairfield, 1979; Browett et al., 2017).

### 3.3 Neutral Sheet Crossing

At a bit further distances, BepiColombo encountered the neutral sheet. As can be seen in Fig. 5, at $\sim 05:25$ UT $|B_{\mathrm{x}}|$ starts to decrease again ($B_{\mathrm{x}} \to 0$ nT) and after $\sim 05:33$ UT $B_{\mathrm{y}}$ also starts to decrease, to end up at a minimum $B_{\mathrm{m}} \approx 3$ nT around 05:43 UT, where then $B_{\mathrm{z}}$ is the dominant component for a short period of time, see Fig. 6. After 05:45 UT, there is a drastic change in the cone angle from $\sim 90°$ to $\sim 180°$ as well as large oscillations in $B_{\mathrm{x}}$, $B_{\mathrm{m}}$ and in the clock angle that varies between $\sim 180$ and $\sim 0°$. There are three of these oscillations, which then are followed by possible crossings of the neutral sheet between 06:08 and 06:15 UT. These neutral sheet crossings are marked by blue vertical lines in Fig. 6, and are seen as $B_{\mathrm{m}}$ reaching $\approx 0°$ nT twice, and the cone angle varying from $\approx 150°$ to $\approx 15°$.

### 3.4 Magnetotail Flapping

Between 06:46 and 07:01 UT there are multiple crossings of $B_{\mathrm{x}} = 0$ nT, with $B_{\mathrm{y}} \approx -4$ nT and a negligible $B_{\mathrm{z}}$ (see Fig. 7). This behaviour is reminiscent of magnetotail flapping observed at Earth (Sergeev et al., 2003), and also evidenced in the Hermean magnetotail (Poh et al., 2020).

At Venus, this phenomenon has also been observed by Rong et al. (2015), with a period of $\sim 3$ min, which is much shorter than the $\sim 7$ min period seen in Fig. 7.

One of the characteristics of flapping is that for consecutive crossings of $B_{\mathrm{x}} = 0$ nT the normal of the current sheet oscillates in the $Y - Z$ plane. We have performed a minimum variance analysis on the four crossings to determine the normal direction to the current sheet. The results are shown in Table 2. The determination of the direction normal to the current sheet appears robust, with eigenvalues well separated for each case and $\lambda_{\mathrm{max}} \gg \lambda_{\mathrm{int}} \gg \lambda_{\mathrm{min}}$. As can be seen, the normal is mainly in the



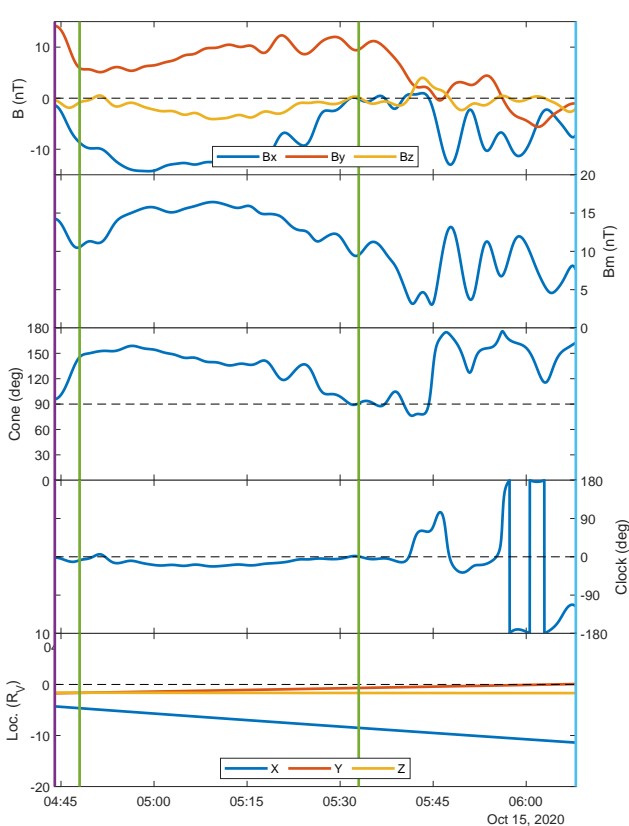

**Figure 5.** Zoom in on time interval (green) that BepiColombo is in the magnetotail proper. This interval shows a strongly draped field with $\theta_x \approx 150°$, with a strong $B_y$ component.



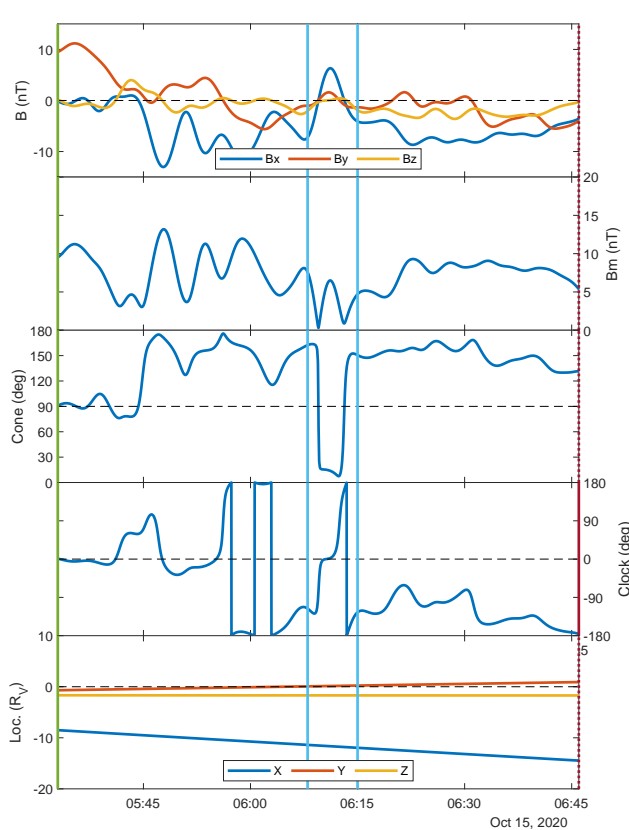

**Figure 6.** Zoom in on neutral sheet crossings interval (blue) Strong oscillations of the field also occur before the selected interval, without crossing $B_x = 0$ nT.





**Table 2.** Minimum variance direction for the $B_x = 0$ nT crossings (cr) rotated such that $n_x > 0$ and the eigenvalues of the MVA, where the ratio $\lambda_{\mathrm{int}}/\lambda_{\mathrm{min}}$ shows that the MVA is well determined.

|  | cr1 | cr2 | cr3 | cr4 |
|---|---|---|---|---|
| start | 06:46 | 06:50 | 06:54 | 06:59 |
| end | 06:49 | 06:53 | 06:57 | 07:01 |
| $n_x$ | 0.17 | 0.16 | 0.10 | 0.26 |
| $n_y$ | 0.66 | 0.12 | 0.93 | 0.52 |
| $n_z$ | -0.74 | -0.98 | 0.34 | -0.81 |
| $\lambda_{\mathrm{min}}$ | $1 \times 10^{-4}$ | $2 \times 10^{-5}$ | $1 \times 10^{-3}$ | $1 \times 10^{-4}$ |
| $\lambda_{\mathrm{int}}$ | $5 \times 10^{-2}$ | $6 \times 10^{-4}$ | $6 \times 10^{-3}$ | $8 \times 10^{-3}$ |
| $\lambda_{\mathrm{max}}$ | 3 | 3 | 3 | 1 |
| $\lambda_{\mathrm{int}}/\lambda_{\mathrm{min}}$ | 500 | 30 | 6 | 80 |

$Y - Z$ plane. For flapping, one would expect then that for $n_y > 0$ there is an alternately positive and negative value for $n_z$. This is only the case for the last three crossings.

After these multiple crossings of $B_x = 0$ nT, there are two more excursions from one lobe to another, and then at $\sim 07:45$ UT the magnetic field strength basically arrives at a more-or-less constant value of $B_m \approx 5$ nT (see Fig. 3). The cone angle slowly rotates from $\theta_c \approx 20°$ to $\theta_c \approx 140°$ and then rotates back again, which is a characteristic as well of flapping activity within the tail. The flapping period is about 7 minutes.

### 3.5 Exiting the bow shock

As BepiColombo continues its path down and across the tail, it will eventually encounter the bow shock/wave again. It is not clear what this structure may look like so far down the tail, and thus from the magnetometer data it is difficult to determine where this crossing happened.

In order to get an estimate of where the crossing could have happened, we determine where the cone angle of the magnetic field varies around the average Parker-spiral angle of $\theta_P \approx 36°$. This occurs around $\sim$14:00 UT at a distance of $X_{\mathrm{VSO}} \approx 48\,R_V$ and $Y_{\mathrm{VSO}} \approx 11\,R_V$.

## 4 Plasma Data

### 4.1 BepiColombo

Fig. 8 shows PICAM, MPO-MAG and MEA1 observations. The PICAM data in Fig. 8 (first panel) show a strong signal of $\sim 1$ keV solar wind ions, both upstream of the bow shock and inside the induced magnetotail. There are variations in the energy of the observed protons $E_p$, especially in the downstream region of the bow shock, between $\sim 04:45$ and $\sim 06:00$ UT with $6 \times 10^2 \leq E_p \leq 2 \times 10^3$ eV. Clear bursts of increased counts occur at the same time as an increase in the MEA electron

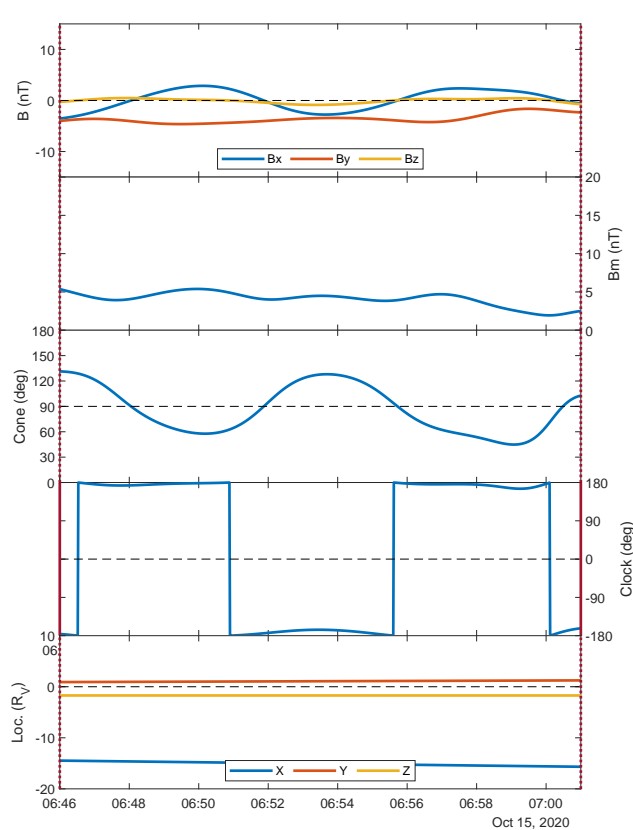

**Figure 7.** Zoom in on magnetotail flapping interval (red). The cone-angle clearly shows alternating magnetic field directions (tailward $\theta_c > 90°$ and Venusward $\theta_c < 90°$). The clock-angle remains basically the same ($\phi_c = 180° \equiv \phi_c = -180°$).





**Table 3.** A comparison of the boundary crossings determined from MEA and from MAG. There are two intervals for the plasma sheet for MAG, the second corresponds to the flapping interval.

| MEA | | crossing | MAG | |
|---|---|---|---|---|
| time in | time out | into/out of | time in | time out |
| 04:08 | 13:04 | Bow Shock | 04:14 | 14:00 (?) |
| 04:53 | 07:37 | Induced Magnetosphere | 04:48 | 07:45 |
| 05:41 | 06:03 | Plasma Sheet | 05:43 | 06:15 |
| - | - | Plasma Sheet | 06:46 | 07:01 |

counts is seen at energies between $\sim 32$ and $\sim 100\,\mathrm{eV}$. There are no measurements of the S/C potential, and therefore one caveat is that the energies for both electrons and ions might have some level of uncertainty. Nevertheless their trend should not

be affected, as the timescale for changes in the S/C potential is often much longer than the time interval of interest.

Interestingly, after crossing the bow shock, there is no clear reduction in the ion energy, which remains at solar wind level until $\sim 05:35$ UT. This is caused by the location of the bow shock crossing at $\sim [-1.5, -2.3, -1.5]\,R_V$, where the reduction of the plasma velocity is much smaller than towards the sub-solar point (see e.g., Spreiter et al., 1966; Spreiter and Stahara, 1994; Schmid et al., 2021). Overall, however, the counts decrease when BepiColombo moves further into the downstream region.

There are a few significant increases in the count rate, marked by grey and blue transparent boxes in Fig. 8. The grey boxes are when the spacecraft has entered the magnetotail, and there does not seem to be a good correlation between the PICAM and MPO-MAG data as far as these increases in counts is concerned. The first one is during a rotation of the field from $B_{\mathrm{x}}$ into $B_{\mathrm{y}}$, the second shows no real peculiarities in the MPO-MAG data and the third shows a strong dip in $B_{\mathrm{x}}$. The lack of one correlation between MPO-MAG and PICAM could be due to the FoV of PICAM which limits its visibility of the whole sky,

so that some ion jets might be missed in observations.

For the blue boxes, which occur during the period where the magnetic field is oscillating strongly, there seems to be a correlation between $B_{\mathrm{m}}$ and the PICAM count rate. The bursts of high counts correlate very well with the decreases in the total magnetic field.

Finally, during the interval labelled as "magnetotail flapping" (red-edged box) there is no increase in electron energy, but

there is a decrease in counts in the low-energy bins below 10 eV.

The MEA1 instrument was turned on from 14 October 2020 03:45:07 UT until 16 October 2020 04:25:51 UT. MEA1 measured low-energy electrons during the flyby except during wheel off-loading. The time-energy spectrogram of electron omnidirectional counts obtained every 4 seconds in the low-resolution telemetry mode is shown in Fig. 8, last panel. Table 3 shows the crossing times of the various regions as deduced from MEA1 data.. The times for some of the crossings are slightly

off with respect to the magnetometer data (see also Table 1).

– The purple box is identified as the magnetosheath where the magnetic field was mainly in the $Y_{\mathrm{VSO}}$-direction. The MEA data show after $04:08$ UT a different population of electrons than that of the solar wind. The main population

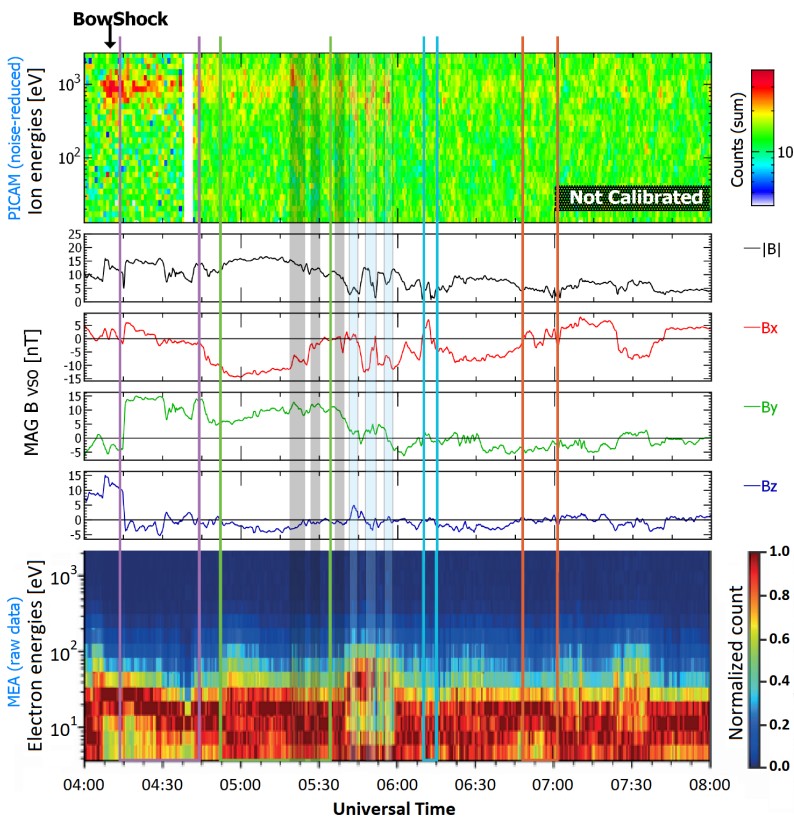

**Figure 8.** Four hours of the Venus 1 flyby as seen by different instruments. Top panel shows the PICAM data, where the $\sim 1\,$keV protons are clearly visible before and behind the bow shock. The data gap near $\sim 04:40$ UT is caused by a mode change. The middle four panels show the MAG data, B-field magnitude and components.. The bottom panel show the linearly normalized MEA omnidirectional electron count time-energy spectrogram. The coloured boxes (purple, green, transparent grey, transparent blue, cyan and red) show the different intervals as defined in the text.

    is at energies between $20\,$eV and $40\,$eV, with some variation to lower energies as the spacecraft moved deeper into the magnetosheath.

– The green box is identified as the magnetotail, where the cone angle is $\theta_c \approx 150°$. In the MEA data this region shows a much broader distribution of electron energies between $3\,$eV and $40\,$eV. Between the purple and green box a magnetic field rotation takes place from $\theta_c \approx 90°$ to $\theta_c \approx 150°$. Already before the end of the purple box the electron energy distribution starts to broaden and reaches a maximum energy width at $04:53$ UT, later than the rotation of the magnetic field. At the end of the green box, the spacecraft seems to move near the neutral sheet, with $B_m < 3\,$nT, followed by

strong oscillations in $B_x$ and $B_m$. The MEA data show two electron populations, one at $3-10\,$eV and one at $30-80\,$eV.




This splitting of the electron population can be caused by acceleration through the electric field generated by the magnetic field gradients.

- The blue box shows the actual crossing of a neutral sheet, with $B_\mathrm{m} \approx 0\,\mathrm{nT}$. Interestingly, there is no signature in the MEA data here, just a broad energy distribution of the electrons.

- The red box is the location of the "magnetotail flapping", the multiple crossings of $B_\mathrm{x} = 0\,\mathrm{nT}$. The MEA data show a decrease in the low-energy electron population.

## 4.2    Comparison to a Venus Express magnetotail flapping event

Venus Express (VEX, Svedhem et al., 2007) also observed magnetotail flapping in the near-Venus tail around $\sim (1.5, 0.1, 0.5) R_\mathrm{V}$ on 24 November 2007, as shown in Fig. 9, discussed by Rong et al. (2015). These authors stated that, different from the Earth's

magnetotail where the source for flapping is expected at the centre (Sergeev et al., 2003; Davey et al., 2012), the source for the flapping in Venus's tail is located near the boundaries between magnetotail current sheet and magnetosheath.

The second and third panels show the magnetometer data (Zhang et al., 2006) $B_\mathrm{x,y,z}$ and $B_\mathrm{m}$. The first and fourth panels show the Analyser of Space Plasma and Energetic Atoms, ASPERA-4-IMA ion mass composition sensor derived proton differential flux at a time resolution of 12 s and the ASPERA-4-ELS electron sensor derived electron differential flux time-

energy spectrogram at a time resolution of 4 s (Barabash et al., 2007). The black (white) vertical dotted lines show where $B_\mathrm{x} = 0$ nT, the two magenta vertical dotted lines show where VEX approaches $B_\mathrm{x} = 0$ nT, but does not cross over.

The IMA time-energy spectrogram shows some weak bursts at $\gtrsim 1$ keV (most likely solar wind protons). At lower energies, protons between $\sim 20$ and $\sim 200$ eV, there are three bursts in the time-energy spectrogram. These seem to be correlated with VEX being in the lobes of the magnetotail, at $|B_\mathrm{x}| \approx 20$ nT. This is different from what was observed by BepiColombo, where

the PICAM bursts seemed to be correlated with minimal observed magnetic field strength.

The ELS spectrogram shows that when the spacecraft approaches the centre of the tail, the flux at higher energies increases (near the vertical lines in Fig. 9) indicating that there are more energetic electrons in the central plasma sheet of Venus's induced magnetotail. Note that the flux is strongly reduced when VEX is in the lobes, whenever $|B_\mathrm{x}|$ increases, indicating that the energetic electrons are a feature of the central plasma sheet.

There are, however, clear differences between the flapping events as observed by BepiColombo and VEX. First of all, the flapping amplitude is about twice as large for VEX. Secondly, in the blue boxes in Fig. 8, where we see the splitting of the electron populations, there are no $B_\mathrm{x} = 0$ nT crossings, so BepiColombo remains in one lobe of the induced magnetotail. Later, in the red box, BepiColombo does cross $B_\mathrm{x} = 0$ nT multiple times.

Comparing the electron energy distribution for MEA1 and ELS for the blue boxes in Fig. 8 and Fig. 9 one can see that for

ELS there is no splitting of the electron population into two energy bands. Also ELS does not show a drop in flux at the $B_\mathrm{x} = 0$ nT crossing as is observed in the MEA1 data in the red box.

Naturally, these differences may well be caused by the difference in location of the spacecraft with VEX near $X \approx 1.5 R_\mathrm{V}$ and BepiColombo near $X \approx 15 R_\mathrm{V}$, as well as the different spacecraft speeds and instrument cadences.

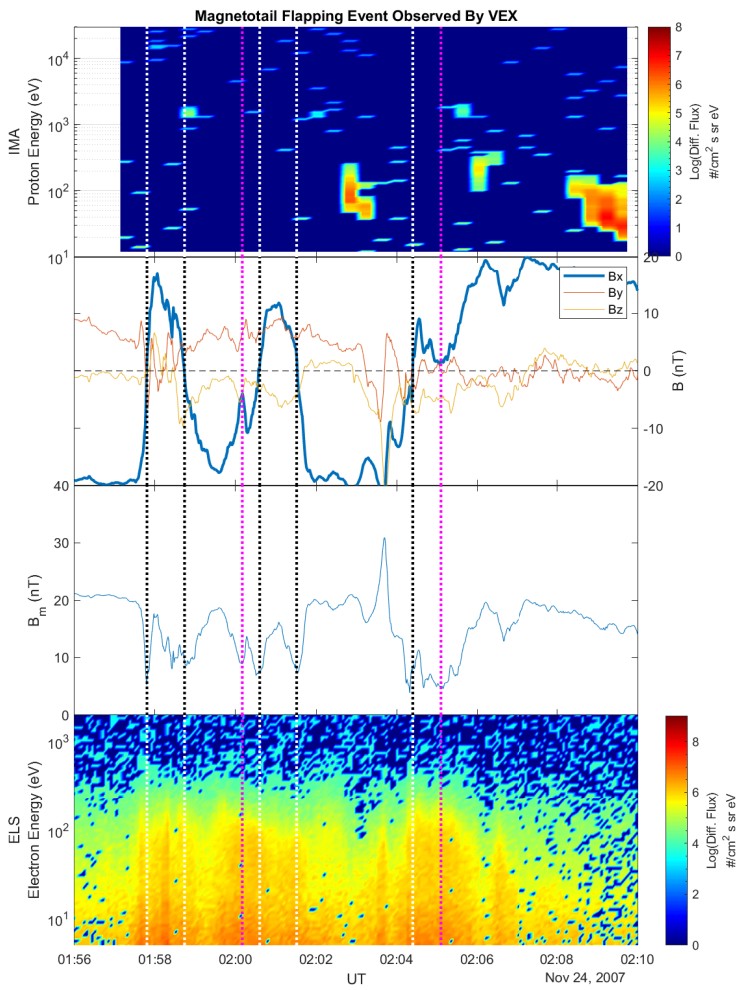

**Figure 9.** Magnetotail flapping event at Venus in the near-tail (near $\sim (1.5, 0.1, 0.5)$ observed by Venus Express on 24 November 2007 (see also Rong et al., 2015). The top panel shows the proton differential flux time-energy spectrogram. The second panel show the three components of the magnetic field. $B_x$ is in a thick blue line showing how it oscillates over the spacecraft moving VEX from one lobe to the other. The third panel shows $B_m$ with clear dips when $B_x = 0$ nT. The bottom panel shows the electron differential flux time-energy spectrogram. The white/black vertical dotted lines show the $B_x = 0$ nT crossings, where magenta lines show times when $B_x$ approaches but does not reach 0 nT


### 4.3 Mariner 10, Galileo and Pioneer Venus

As Mariner 10 was on its way to Mercury, it passed through Venus's magnetotail (wake) on 5 February 1974, in a similar orbit in the $X - R$-plane as BepiColombo, but at positive $Z$ (see, Lepping and Behannon, 1978). The wake magnetic field data were studied starting at a distance of $\sim 100 R_V$ from the planet. No evidence was found for a bow shock crossing entering the far tail. No typical magnetotail structure (as compared to Earth) was observed, but the authors found that, when the magnetic field direction and the spacecraft velocity vector aligned, the direction was not predominantly along $X_{VSO}$, as one would expect for

a magnetotail along the orbit of Mariner 10.

The data were categorized into three bins, quiet, disturbed and mixed. The longitudinal (cylindrical) component of the field was observed to rotate clockwise, when the spacecraft crossed from a quiet to a disturbed region. Lepping and Behannon (1978) interpreted this as Mariner 10 entering into planetary magnetotail, however, with a significant $Y$-component for most crossings. The IMF, after Mariner 10 crossed the bow shock, exiting the induced magnetosphere was $F = 20\gamma$, $\phi \approx 360°$ and

$\theta \approx 0°$ (Ness et al., 1974). This means a mostly radial magnetic field. These multiple crossings do not seem to happen for BepiColombo. During the Solar Orbiter flyby such crossings were observed (Volwerk et al., 2021).

The Galileo spacecraft used a Venus gravitational assist on its way to Jupiter. During this Venus flyby, the orbit skimmed the bow shock. Kivelson et al. (see e.g., 1991) used the magnetometer data to investigate the cross section of the bow shock. They found that it seemed to be smaller when its direction was aligned with the IMF when compared to when its direction was

perpendicular to the IMF.

Using Pioneer Venus data, Slavin et al. (1989) studied Venus's near-tail region, at $|X| \lesssim 12\ R_V$. Twelve passes through the central induced magnetotail (in the period 1981 – 1983) along Pioneer Venus's polar orbit were studied, and it was found that the spacecraft traversed the central current sheet multiple times during each crossing. The quasi-period of the crossings is $\lesssim 10$ min (as estimated from their figures). The spacecraft moved between clearly defined oppositely directed fields. This behaviour

could be considered magnetotail flapping, however, this was not yet a named phenomenon.

The main difference between the BepiColombo passage through Venus's tail and the orbits studied by Slavin et al. (1989) is the IMF direction. For BepiColombo the IMF is mainly in the $Z$-direction, whereas for the Pioneer Venus events the IMF is mainly in the $Y$-direction. This means that the morphology of the induced magnetotail is rather different. For the Pioneer Venus orbits the central current sheet was almost in the $X - Z$-plane d, whereas for BepiColombo the central current sheet is

almost in the $X - Z$-plane.

## 5 Solar Wind Interaction: Context for Venus's magnetotail observations

### 5.1 BepiColombo Solar Wind Conditions

Many of the features seen in the magnetometer data are in good agreement with what one would expect for draped magnetic field lines from the solar wind inducing a magnetosphere around Venus. However, it is also clear that the solar wind was

disturbed based on the significant activity of the tail such as the multiple neutral sheet crossings and flapping of the tail.

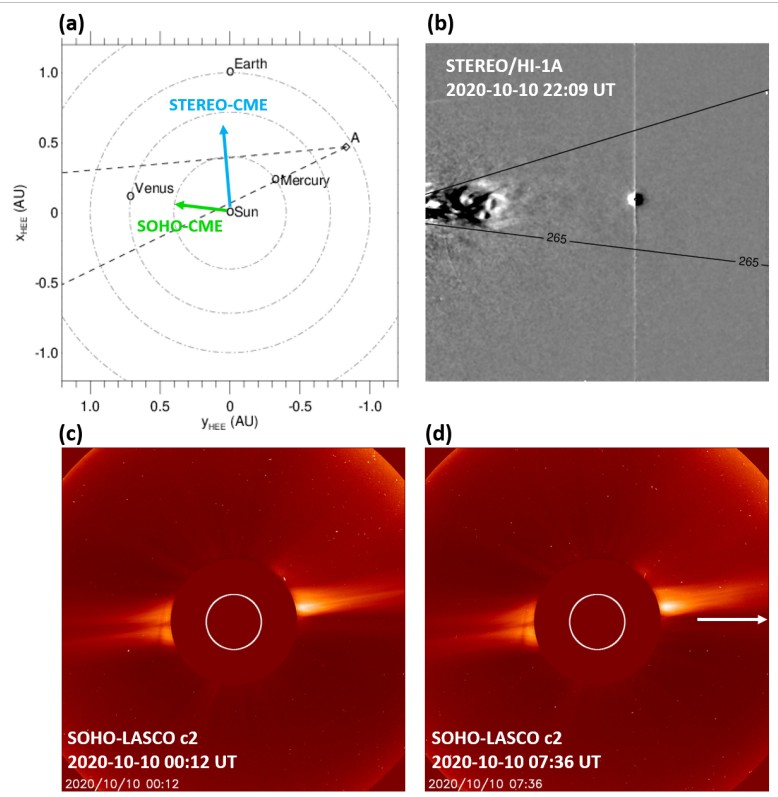

**Figure 10.** STEREO and SOHO observations. (a) Planet and STEREO A (A) positions and elongation range of the field-of-view in the plane of the sky of STEREO-A/HI1 (this is a 20 degree angle in the two dimensions). The blue and green arrows indicates the direction of the nose of the CMEs seen by SOHO and STEREO nearly simultaneously. (b) STEREO-A/HI1 image in a running difference format (where, in each case, the previous image is subtracted from the current image to highlight changes). The most northern and southern position angles of the CME spans are plotted as black lines and Venus is the bright dot towards the center. (c and d) SOHO-LASCO c2 images before and during the CME transit, respectively. The white arrow indicates the plasma motion off the west limb, which is more evident at the movie created at the SOHO Movie Theater (https://soho.nascom.nasa.gov/data/Theater/).

In a pre-flyby study, McKenna-Lawlor et al. (2018) discussed the space weather near Venus during the BepiColombo flybys, where one could expect interactions with, e.g., Interplanetary Coronal Mass Ejections (ICMEs) and Corotating Interaction Regions (CIRs). However, during the actual BepiColombo flyby 1 our interpretation of the solar wind interaction with the induced magnetosphere is hampered by the lack of an upstream solar wind monitor. Nevertheless, thanks to pre- and post-flyby observations made by the MPO-MAG, BERM and MEA1 instruments, together with observations made by STEREO-A, SOHO and a solar wind numerical simulation, the space weather context of the encounter is reconstructed.

Fig. 10 shows some STEREO and SOHO observations of a potential CME that may have hit Venus during the BepiColombo flyby. The Space Weather Database Of Notifications, Knowledge, Information (DONKI) catalogue reports a single CME event



for the period 7-13 October, which corresponds to the time needed for a CME to reach Venus by the time of the BepiColombo

flyby. The DONKI catalogue indicates that this event was observed by SOHO-LASCO c2, c3 and STEREO-A SECCHI instruments with a starting time on 10 October 2020 04:24 UT and a speed of 270.0 km/s. This specific DONKI run also lists a direction right on the western limb (with respect to Earth). Moreover, no clear source eruption (filaments or prominence activity) in the Solar Dynamic Observatory (SDO) imagery was observed. This event is clearly seen by the SOHO-LASCO instrument in panels c and d, where plasma outflows along a west limb streamer seen from L1 go into Venus's direction (Venus

was near in quadrature with respect to Earth, see panel a).

However, the Heliospheric Cataloguing, Analysis and Techniques Service (HELCATS), which provides the official interplanetary CME catalogue of the STEREO HI instruments, reveals a rather different scenario. HELCATS (Harrison et al., 2018) catalogues the derivation of CME kinematic properties (direction, speed and launch time) from geometric fitting techniques applied to the HI observations, described in Davies et al. (2013). The technique applies a Self Similar Expansion (SSE) approach

that assumes a circular CME topology, expanding within two fixed position angles. The validity of this assumption depends on the nature of the particular event under study and the derived parameters should be regarded as best estimates in the spirit of that assumption. For the event under study (Figure 10), the SSE fit suggests that the HI-observed CME was a weak CME ejected by the Sun on 9 October at 23:05 UT with a speed of 283 km/s. The first observation of the CME by the HI1 camera was on 10 October at 10:09UT and the fit indicates that it was near Earth-directed. No clear arrival was detected in the vicinity

of Earth, though the solar wind parameters do show some signs of disturbance, though, of course, the CME might have simply passed near to the Earth. However, overall, we conclude that there is some evidence that the near simultaneous weak CMEs observed by both STEREO-A and SOHO were not the same, that is, we witness a near-Earth directed CME from the STEREO instrumentation and coincident CME activity associated with a streamer with the SOHO data, that is directed towards Venus. The SOHO-observed CME could well have hit BepiColombo at thetime of its first flyby to Venus.

In order to estimate the arrival time of this CME at Venus, Fig. 11 shows the main outputs of a SUSANOO simulation. Panels a-c show three stills of the solar wind velocity in the ecliptic plane. For completeness, the simulation has also included the following CME observed by SOHO-LASCO, that was ejected on 13 October 2020 at 21:12 UT with close direction to Venus as well. Panels d-e show the same simulation in a 3D view. Finally, panel f shows the IMF (magnitude), the density, and the velocity of the solar wind temporal variation. The arrival and ending time of the CME transits at BepiColombo is

indicated with vertical dashed lines in panel f. According to the simulation, the CME arrived at BepiColombo and Venus on 13 October 2020 at about noon (UT time). Since the velocity was relatively low, it needed a couple of days to transit Venus. The simulation predicts that the CME left Venus on 15 October 2020 at about 15 UT. Therefore, the BepiColombo flyby most probably occurred while the CME was still transiting Venus. The second CME most probably also hit Venus after the flyby arriving on 17 October 2020 as predicted by the simulation.

Figure 12 shows the actual solar wind observations made by BepiColombo from 12 to 17 October. In particular, it shows: the IMF measured by the MPO-MAG in panel (a), the proxy for galactic cosmic ray flux measured by BERM in panel (b), and the solar wind energetic electron spectra from MEA1 in panel (c). The reason for the large data gaps in panel (c) is that MEA1 only operated for a few hours around the closest approach. This figure corroborates that the solar wind was indeed




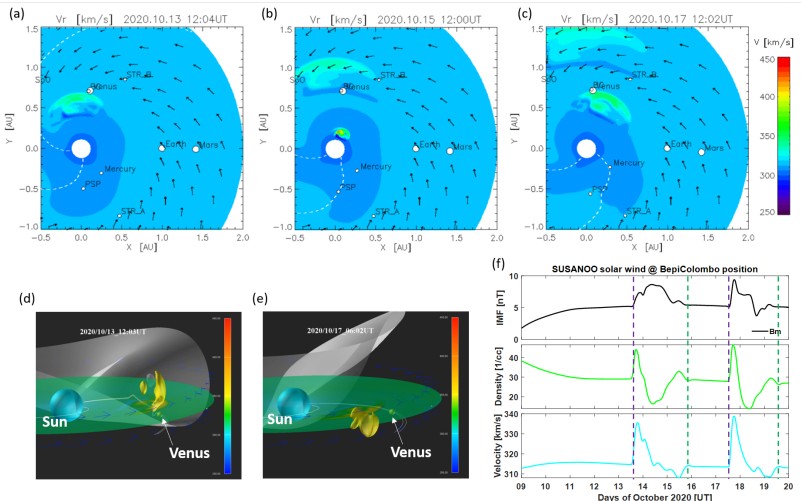

**Figure 11.** Stills from the SUSANOO simulation performed for the period 9-18 Ocotber 2020. (a-c) 2D plots of the solar wind velocity at the ecliptic plane on 13 October 2020 at 12:04 UT (CME arrival at BepiColombo), 15 October 2020 at 12:00 UT (trailing edge of the CME leaves BepiColombo), and 17 October 2020 at 12:02 UT (a second CME arrival at BepiColombo). The Sun is the white largest circle, the different planets and satellites are also labelled and the black arrows show the direction of the solar wind flow. The greenish blob toward Venus represent the CMEs of this study. The background colours represent the speed, as indicated in the colour scale. (d-e) Same as before but in 3D. The green plane is the ecliptic and the white-transparent surface is the heliospheric current sheet. (f) Time series of IMF (magnitude in black), density (in green) and velocity (in light blue) at the BepiColombo location obtained from the simulation. The vertical dashed purple and green lines indicate the arrival and end time of the CMEs at BepiColombo.

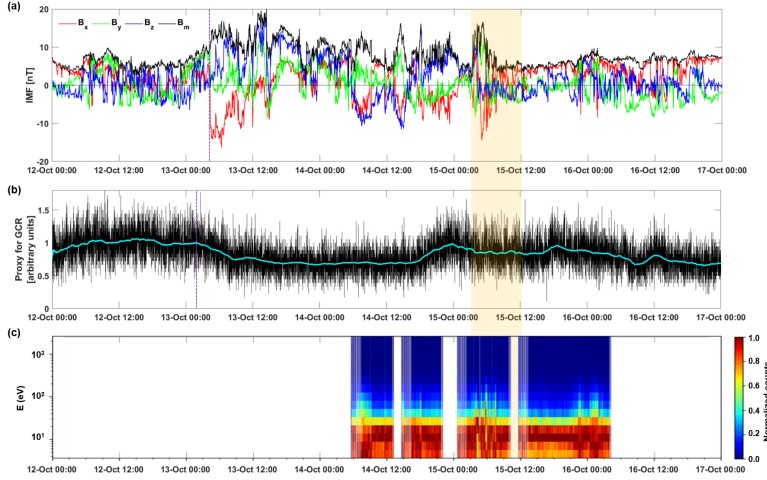

**Figure 12.** Solar wind observations: (a) Magnetic field observations from MPO-MAG in VSO, (b) Proxy for Galactic Cosmic Rays (GCR) observations from BERM, (c) energy electron spectrograms from MEA1. The arrival time of the coronal mass ejection (CME) is marked with a vertical purple dashed line and the Venus transit (same period of the observations of this paper) is marked with a yellowish box.





clearly disturbed. The overall magnitude of the solar wind is $\sim$15 nT for most of the period, similar to the induced magnetic

field values observed at Venus (yellowish box). The CME arrived at BepiColombo on 13 October at 04:20 UT (vertical purple dashed line), where a moderate rise in the IMF magnitude from 10 to 15 nT was observed simultaneously with a rotation in the three components of the field, mainly seen in $B_x$. Moreover, starting a few hours before, a significant reduction in the Galactic Cosmic Ray (GCR) flux was observed (purple dashed line in panel b) and the GCR flux remained low for almost 2 days. These kinds of reductions followed by a gradual recovery could be associated with Forbush decreases, which are produced by the

magnetic flux rope inside the CME that scatters away the incoming GCR (Witasse et al., 2017). In this sense, Forbush decreases are good indicators of CME arrivals. The IMF magnitude was maintained at $\sim$15 nT and the level of GCR was kept relatively constant until few hours before the encounter with Venus. MEA1 observations also agree with the idea of a CME transiting as the variability observed in the solar wind energetic electron observations matches very well with the magnetic variability, especially when the $B_x$ and $B_z$ components are negative. This corroborates the idea that the solar wind was disturbed just

a few hours before BepiColombo's Venus encounter. These small and slow CMEs are transients often seen during low solar activity phases of the solar cycle and are often called stealth-CMEs because, as in this case, no clear source is identified. BepiColombo has already encountered several transient structures of this type, such as that presented in Heyner et al. (2021). Although stealth-CMEs typically are pushed by the solar wind, they have the capacity of interacting with planet's conductive surface and ionosphere plasma: this is especially the case for unmagnetised planets as demonstrated in this study at Venus and

also with similar events at Mars by Sánchez-Cano et al. (2017) and Kajdič et al. (2021).

Right after the flyby, there was still significant solar wind variability and according to the simulation in Fig. 11, these perturbations are most probably caused by the trailing edge of the CME. This means that during the flyby to Venus, the system was most probably immersed in the CME. The MEA1 observations also corroborate this finding showing a large perturbation in the solar wind electrons on 15 October at 22:00 UT at the same time that a large variability is observed on the MPO-MAG

data with $B_x$ and $B_z$ IMF rotations.

## 5.2  Venus Express Solar Wind Conditions

Interestingly, Kajdič et al. (2021) noticed that in November 2007 there was a good alignment of Mercury, Venus, Earth and Mars. In the period of 20 to 27 November, during the presented VEX event, a CME and two stream interaction regions (SIRs) were observed by ACE, STEREO A and B, and Mars Express. With the aforementioned alignment of the planets it stands to

reason that Venus's induced magnetosphere was also impacted by these structures.

This means that the solar wind conditions during the VEX event, shown in Fig. 9, are very comparable with those during the BepiColombo flyby. Even though VEX traversed the tail much closer to the planet, the disturbance of the magnetosphere through an outside source is clear through the flapping motion, driven from outside-in (Rong et al., 2015).



# 6 Conclusions

The first Venus encounter by BepiColombo has shown a new view of the Venusian induced magnetosphere up to about 48 Venus radii downstream. Until this flyby only one spacecraft had ever ventured this far down the magnetotail, Mariner 10 (Lepping and Behannon, 1978). A few months later, in December 2020, Solar Orbiter also had its Venus flyby over a similar distance along the tail investigating its dynamics (see e.g. Fig. 13 and Volwerk et al., 2021).

The asymmetric draping of the magnetic field, just behind the bow shock in the magnetosheath, as observed by Delva et al.
(2017) and modelled by Jarvinen et al. (2013), was confirmed by this flyby. The field pointed in the direction perpendicular to the Venus-Sun line before the spacecraft entered the magnetotail proper.

The magnetotail was very active, with strong oscillations of the magnetic field with $\Delta B/B \approx 0.6$, with a period of $\sim 7$ min. This oscillation or flapping of the magnetotail was slower than what was typically measured by VEX in 2007, where Rong et al. (2015) determined a period of $\sim 3$ min. However, observations in the Earth's magnetotail show that the magnetotail flapping
period varies from $\sim 3$ min. (Sergeev et al., 2003) to $\sim 20$ min. (Zhang et al., 2005).

During the strong oscillations of the magnetic field PICAM measured increased ion fluxes when the total field was at a minimum. At the same time MEA showed that there were two populations of electrons, one below $10\,\mathrm{eV}$ and one between 32 and $100\,\mathrm{eV}$. The latter "hot" population was also observed by Venus Express much closer to the planet.

Despite the low solar activity conditions, the flyby was affected by the impact of a stealth coronal mass ejection that was
travelling at approximately the same speed than the background solar wind and impacted Venus and BepiColombo about 2 days before the closest approach on 13 October. Due to the low speed of this CME, in-situ magnetic and particle observations together with a solar wind simulation indicate that the trailing part of the CME was still affecting Venus at the time of the BepiColombo's closest approach and tail transit. A second CME may have hit both Venus and BepiColombo on 17 October, in principle, not affecting the flyby. Therefore, the highly dynamic tail observed by BepiColombo may be the consequence of
space weather activity.

On 10 August 2021, the second Venus flyby will take place, where BepiColombo will approach the planet from the tail side, and pass closely by the planet in the dayside magnetosphere, as shown in Fig. 13 bottom panel. One day earlier, on 9 August, Solar Orbiter will also have its second Venus flyby in a more similar orbit as the first flyby. This means that both spacecraft can act as solar wind monitors for the other mission, during these flybys. This will be an unprecedented occasion to obtain
two-point global measurements around Venus.

*Data availability.* The BepiColombo MPO-MAG, PICAM, MEA and BERM data, as welll as the Venus Express MAG and ASPERA-4 data are available through ESA's Planetary Science Archive (PSA, https://archives.esac.esa.int/psa). The STEREO data are available through the HELCATS catalogue (https://www.helcats-fp7.eu/catalogues/event_page.html). The SOHO data are available through NASA's SOHO website (https://soho.nascom.nasa.gov/data/Theater/).

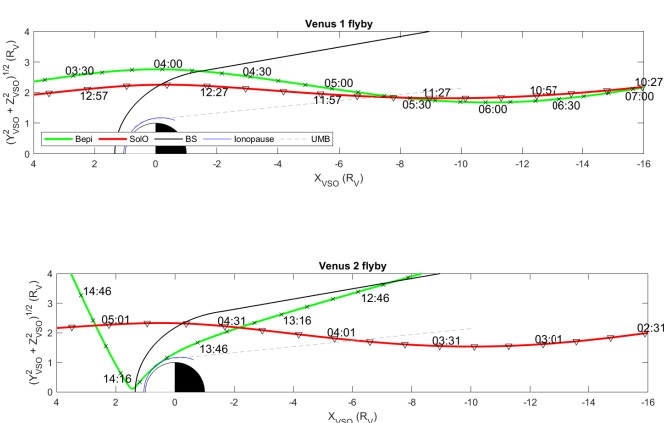

**Figure 13.** A comparison of the first and second Venus flybys by BepiColombo (green) and Solar Orbiter (red). Venus 1 flyby was on 15 October 2020 for BepiColombo and on 27 December for Solar Orbiter. The Venus 2 flybys will take place on 9 August 2021 for Solar Orbiter and on 10 August for BepiColombo.

*Author contributions.* M.V., B.S.-C. and D.H. instigated the MPO-MAG data investigation. J.M. and I.R. calibrated the MPO-MAG data. S.A. and N.A. provided the MEA data and interpretation. A.V., H.J. and G.L. provided the PICAM data and interpretation. Y.M., I.K., D.S. and Y.S. provided the SUSANOO simulations. R.H. provided the STEREO data and interpretation. F.P., D.S., C.S.W., R.N. and W.B. helped with interpreting the results. S.R.M. and Y.F provided the Venus Express ASPERA-4 data.

*Competing interests.* The authors have no competing interests.

*Acknowledgements.* B.S.-C. acknowledges support through UK-STFC grants ST/S000429/1 and ST/V000209/1. C.S.W. is supported by the Austrian Science Fund (FWF) under project N32035-N36. D.H. was supported by the German Ministerium für Wirtschaft und Energie and the German Zentrum für Luft- und Raumfahrt under contract 50 QW 1501. N.A. and S.A. acknowledge the support of CNES for the BepiColombo mission. MEA data analysis was performed with the CL software developed by Emmanuel Penou at IRAP and the AMDA science analysis system provided by the Centre de Données de la Physique des Plasmas (CDPP) supported by CNRS, CNES, Observatoire
de Paris and Université Paul Sabatier, Toulouse. D.S. and D.F. work is financially supported by the Austrian Research Promotion Agency (FFG) ASAP MERMAG-4 under contract 865967. S.R.M. was funded by the Swedish National Space Agency under contracts 79/19 and 145/19.





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
