# Peer review of "Venus's induced magnetosphere during active solar wind conditions at BepiColombo's Venus 1 flyby"

_Annales Geophysicae, 2021_

## Author Comment (AC1)

**Replies to the Referee 1**

'RC1: 'Comment on angeo-2021-24', Anonymous Referee 1, 18 May 2021

**General Statement:** This manuscript summarizes the particles and fields measurements and initial results returned by the BepiColombo Mercury mission during its first Venus flyby (VFB-1) on 15 October 2021. While nearly all aspects of such flybys are driven by requirements related to the spacecraft's safety and timely arrival at their primary destinations, these events constitute special opportunities that have produced important "bonus" science on previous missions. The manuscript is well-constructed and the writing is quite good. All key aspects of the VFB-1 operations and instrument performance are well-documented. New science results from the initial analyses of these measurements indicate that Venus' draped magnetic field tail extends at least 48 Rv downstream of the planet and that the period for tail "flapping" is much broader than previously observed, at least 3 to 7 min. Some minor suggestions are provided below, however, the manuscript reports important new observations of the solar wind interaction with Venus and significant new science results. Further, the BepiColombo VFB-1 data set is documented in detail for future scientific studies to follow. Accordingly, I recommend that the manuscript be published with only minor revisions.

**Specific Comments and Suggestions: Lines 41-49:** I think you should not spend too much time on history, but first surveys of the Venus magnetotail were carried out by the Venera 9 and 10 orbiters in 1975-1976 (for details see Verigin et al., Plasma analysis, JGR, August, 1978; Eroshenko et al., induced magnetic tail, Cosmic Research, 17, 17, 1979). This is very near the time of Mariner 10 primarily magnetosheath flyby, but for the sake of completeness you might consider referencing Venera and 9 and 10's historic contribution.

> Answer: For completeness we have mentioned the Venera 9 and 10 orbiters in the text here.

**Section 2 "The Data":** The description of the BepiColombo mission and the impact of the stacked science spacecraft and SEP carrier cruise configuration may be too brief for Readers who are not already familiar with the mission. You do note the impact of the cruise configuration on the field-of-view of some individual particle instruments in isolated sentences later in the text. However, I would recommend at least a brief overview of BepiColombo's cruise configuration (e.g., MMO behind heat shielding; MPO MAG further from "SEP carrier module" but still seeing some stray B-field contamination) early in Section 2 to provide context for the instrumental considerations that follow.

> Answer: We have added a short paragraph at the beginning of Section 2 to describe the cruise phase formation of the spacecraft, indicating that instruments can be impeded in their performance.

**Section 3.2:** The smoothed magnetic field data and the limited number of cross-tail current sheet crossings may preclude this analysis, but did you examine the angular rotation of B as BC traversed the cross-tail current sheet,? If you did, then were the rotations ¡ 180 deg and, if so, by how much? After the draped IMF flux tubes that make up the "induced" tail slip about the Venus ionosphere and move downstream, they start to "unkink" as the ionospheric plasma and pickup ions from the dayside and flank interaction regions are accelerated by the J x B (aka "magnetic sling-shot" effect – just like in a comet tail) in the cross-tail current sheet. This effect was observed

very clearly in the Pioneer Venus Orbiter data with the magnetic field rotations across the current sheet decreasing (i.e. increasingly below 180 deg) as the downtail distance grew and the speed of the O+ in the cross-tail current sheet increased toward solar wind speeds (Slavin et al., JGR, 1989). Given that, as you point out, the BepiColombo VFB1 Tail encounters were further downtail than the PVO sampling a measurement of the magnetic field rotation across the current sheet crossings would be of great interest, if it is possible with these data?.

Answer: We would like to thank the referee for this interesting question. We know that the magnetotail is flapping, and thus there should be a possibility to check this "unkinking" of the field lines in the tail. We should keep in mind though that the spacecraft remains at $Z_{\mathrm{VSO}} \approx 1$ whilst crossing the tail in the $Y_{\mathrm{VSO}}$-direction. This means that only large oscillations of the tail will probably make the spacecraft cross from one lobe to the other, something discussed in the oscillations and flapping sections in the text. Nevertheless, we used low-pass filtered data, for periods longer than 30 minutes, to study the large-scale cone angle and looked at large rotations. These were found and indeed there is an indication that the rotation gets smaller as BepiColombo moves farther down the tail, from $\sim 132°$ at X = -15 to $\sim 60°$ at X = -45.

---

## Author Comment (AC2)

**Replies to Comment CC1**

**'CC1: Comment on angeo-2021-24', Olivier Witasse, 06 Jun 2021**

**This is a well written paper. Good to see some results at Venus from Bepi Colombo. I have a few comments:**

**Figure 12 shows data from the radiation monitor. The period of the Venus tranit seems to be characterised by a small drop in the radiation level. it would be interesting to have a comment in the paper. Possibly to be compared to the findings of Honig et al. Ann. Geophys., 37, 903–918, 2019 https://doi.org/10.5194/angeo-37-903-2019, a drop of 8% in the radiation data near comet 67P.**

Answer: Thank you so much for pointing out this decrease. It is difficult to conclude whether this decrease is consequence of a similar mechanism at Venus and comet 67P, or it is an effect of the solid angle. In any case, it is indeed very interesting and timing that similar decreases are observed at both unmagnetised bodies ($\sim$8% at Comet 67P and $\sim$12% at Venus). The following sentence has been added:

*In addition, we also note that during the closest approach (starting and ending right before and after the Venus inbound and outbound, respectively), the radiation monitor BERM detected a moderate reduction of 12% in the GCR flux proxy. The reason for this reduction is unknown and could be consecuence of a solid angle effect from Venus, although we note that the flux level is mantained nearly constant during the flyby at the same time than the BepiColombo-Venus distance changes, and so, the solid angle. Interestingly, Rosetta also saw a similar reduction in the GCR flux of 8% in the vicinity of the comet 67P/Churyumov-Gerasimenko which could not be attribuited to any known mechanisms [Honig et al., 2019] .*

**Can the BERM observations help to figure out when you leave the tail (14:00 ? in the text)?**

Answer: This is an interesting observation. Although the recovery at 14:00 on 15 October could be a sign for exiting the tail, it could also be consequence of Venus blocking less GCR flux as soon as BepiColombo goes further. Since the reason is unkown, we prefer to not make further comments in the manuscript.

**Fig 2: it is not standard to have the Y axis labels either on the left or on the right side in the same figure! The color bar explnation is not given the caption.**

Answer: In order to avoid overlapping Y-axis labels it has become standard to alternatingly have the labeling on the left and the right side of the figure. We have added the description of the colour bar in the caption of the figure.

**Fig 11, d-e: what are the yellow bubbles? Not explained in the caption.**

Answer: This is a small color misunderstanding! We referred to them as "greenish blobs" in the caption. However, it is true that they look more like yellowish. The caption has been updated.

**References**

Honig, T., Witasse, O. G., Evans, H., Nieminen, P., Kuulkers, E., Taylor, M. G. G. T., Heber, B., Guo, J. and Sánchez-Cano, B. [2019], 'Multi-point galactic cosmic ray measurements between 1

and 4.5 au over a full solar cycle', Ann. Geophys. **37**, 903–918.

---

## Author Comment (AC3)

**Replies to the Referee 2**

**'RC2: 'Comment on angeo-2021-24', Anonymous Referee 2, 19 Jul 2021**

——————— **General comments:** The manuscript Venus's induced magnetosphere during active solar wind conditions at BepiColombo's Venus 1 flyby by Volwerk et al. presents highly interesting and unique measurements from Venus' long magnetotail made by the BepiColombo spacecraft. The figures in the paper show the measured magnetic field, ion and electron measurements in a very clear and informative way. Moreover, the authors interpret the data and put the observation into a wider context by discussing and comparing observations with the previous plasma and field observations from the Venus magnetotail. The paper is logically structured, and text is clearly written. In addition to the presentation of the data, the work is valuable also because it is foreseen that in the future the presented observations will motivate global modelling works. Some more details should, however, be provided before the work is ready for publishing, please see below.

——————— **Individual scientific remarks:** * Please describe in more details what can be seen in Figure 11:

**- The authors state that the black arrows show the direction of the magnetic field, but they look rather like directions of the IMF.**

Answer: The black arrows show the direction of the solar wind flow in panels a, b and c, NOT the direction of the magnetic field. This is clearly stated on the caption of the Figure.

**- Fig. 11d) and 11e): Colour bars are hardly visible. Is the yellow region a constant velocity surface? Does the mostly green colour region show the speed of the solar wind on the ecliptic plane?**

Answer: The yellowish blobs toward Venus are the CMEs of this study. The colorbar velocity is only applicatble to the CME. This information is now written in the caption. Also, the colorbar has been improved.

**\* A brief piece of information. The analysed flyby is exciting also because the observations may include effects of an ICME. Interestingly, such a situation when an ICME hits Venus has been analysed, and also simulated, already when the VEX observations has been analysed (Dimmock et al., JGR, 2018).**

Answer: Thank you very much for this reference, which indeed it is very related to this work. A paragraph on this paper has been added to section 4.2.

——————— **Technical corrections/suggestions:** * [Fig. 8] When the authors refer to the time range between purple, green, cyan and red lines show in Fig. 8, they use the term "box" although there are no purple, green, cyan and red boxes but just lines. This terminology could be clearer.

Answer: The text has been changed to "The coloured lines (purple, green, cyan and red) and the shaded areas (transparent grey, transparent blue)"

**\* "ASPERA-4-IMA" is typically written as "ASPERA-4 IMA" and "ASPERA-4-ELS" as "ASPERA-4 ELS".**

Answer: corrected

* [l. 223] Here "...magnetotail flapping in the near-Venus tail around   (1.5,0.1,0.5) RV ..." the x position should probably be "(-1.5,0.1,0.5)" i.e. the X VSO should be negative in order the position would be on the night side.

* [l. 247-248] Similarly, this "...  VEX near X   1.5RV and BepiColombo near X   15 RV,..." should probably read as "...  VEX near X   -1.5 RV and BepiColombo near X -15RV,...". Also, in Fig. 9. figure caption this "...near   (1.5,0.1,0.5) observed..." should probably read as "...near   (-1.5, 0.1, 0.5) observed..."

  Answer: Indeed, of course the minus signs had disappeared

* [l. 259] The value of F would be good to express in SI units, i.e. as "F = 20 nT".

  Answer: this is corrected in the text: "F= 20nT (in the paper denoted as 20 $\gamma$)"

* [l. 309] "thetime" - "the time"

  Answer: Change done.

* [Fig. 11 figure caption] "Ocotber" - "October"

  Answer: Change done.

* [l. 381] "as welll" - "as well"

  Answer: Change done.

---

## Author Response (AR1)

**Replies to Editor after minor revisions**

**Comment: My only additional comment, in response also to one non-reviewer comment in the interactive discussion, is to consider adding a panel in Fig. 12 showing the time series of the solid angle available for SEP/GCR access to BERM/Bepi-Colombo. You may do that in the same sense shown in this paper here: `https://www.sciencedirect.com/science/article/pii/S0019103517305705?via%3Dihub` (see Figure 10 and associated discussion).**

Answer: Thank you so much for this comment. The part not-shadowed by the solid angle has been included in the Figure. We note that we have followed a different approach for this calculation as it was not clear to us how the authors of the the mentioned paper calculated the flux units shown in their Figure 10. Our approach is explained in the text as follows:

*The reason for this reduction is unknown and could be consecuence of a solid angle effect from Venus, similar to those observed at Mars at an orbiter periapsis [e.g. Semkova et al., 2018]. For this reason, panel (b) shows the part of the BepiColombo's solid angle not shadowed by Venus in orange, where 1 stands for null shadowed (null solid angle). The solid angle has been calculated as $\omega = 2\pi(1 - cos\theta)$, where theta is the linear angle between the BepiColombo's distance to the center of the planet and the BepiColombo's distance to the limb of the planet, which in turn is calculated as the arcsine of the ratio between the Venus' radius and the BepiColombo's distance to the center of the planet. The part of space not shadowed by the solid angle (orange line) is calculated as $1 - (\omega/4\pi)$. We note that the BERM flux level reduction that can be attribuited to the solid angle effect lasts much shorter than the actual nearly constant reduction found. Nevertheless, other effects more difficult to discern may be also playing a role, such as shadowing from the own spacecraft due to attitude changes.*

**Comment: That paper also shows short and longer term variations on GCR/SEP flux at Mars (a Venus-type object when it comes to its solar wind interaction), so some comparisons may be drawn with depletions/dropouts you show in your Fig. 12, in addition to the reference to the 67p observations recommended in the interactive discussion.**

Answer: A mention to those dropouts has been done in the text (see previous reply).

**Finally, it may be useful to provide some basic information about BERM - e.g. what is the energy threshold of the measurements shown, e.g. ¿40 MeV, ¿100 MeV - what are the proton gyroradii involved? That may play a role in the solid angle shadowing, too.**

Answer: BERM energy information is now provided in the Data section as well as a reference. This reference is a paper that is currently under review at the BepiColombo's special issue in the journal Space Science Review. We know that it is not the best practice to cite a paper under review, but this is the only reference available at the moment. Regarding the observations in Figure 13, a general sentence about other possible sources has been added (see first reply). However, we would like to not go into more details as BERM is currently being callibrated and the energy limits and capability of each channel are being tested. Despite this, we think it is good to show the observations together with MPO-MAG and MEA1, so other reserarchers can see the full picture taken by BepiColombo.

Many thanks for all your comments!

**References**

Pinto, M., Sanchez-Cano, B., Moissl, R., Cardoso, C., Gonçalves, P., Assis, P., Vainio, R., Oleynik, P., Lehtolainen, A., Grande, M. and McComas, A., [2021], 'The Bepicolombo Radiation Monitor, BERM', Space Sci. Rev. p. Submitted(?).

Semkova, J., Koleva, R., Benghin, V., Dachev, T., Matviichuk, Y., Tomov, B., Krastev, K., Maltchev, S., Dimitrov, P., Mitrofanov, I., Malahov, A., Golovin, D., Mokrousov, M., Sanin, A., Litvak, M., Kozyrev, A., Tretyakov, V., Nikiforov, S., Vostrukhin, A., Fedosov, F., Grebennikova, N., Zelenyi, L., Shurshakov, V. and Drobishev, S. [2018], 'Charged particles radiation measurements with Liulin-MO dosimeter of FREND instrument aboard ExoMars Trace Gas Orbiter during the transit and in high elliptic Mars orbit, ', Icarus **303**, 53 – 66.